# The Acute Effects of a Swimming Session on the Shoulder Rotators Strength and Balance of Age Group Swimmers

**DOI:** 10.3390/ijerph18158109

**Published:** 2021-07-30

**Authors:** Nuno Batalha, Jose A. Parraca, Daniel A. Marinho, Ana Conceição, Hugo Louro, António J. Silva, Mário J. Costa

**Affiliations:** 1Departamento de Desporto e Saúde, Escola de Saúde e Desenvolvimento Humano, Universidade de Évora, 7000-654 Évora, Portugal; jparraca@uevora.pt; 2Comprehensive Health Research Centre (CHRC), Universidade de Évora, 7000-654 Évora, Portugal; 3Department of Sport Sciences, University of Beira Interior, 6201-001 Covilhã, Portugal; marinho.d@gmail.com; 4Research Centre in Sports, Health and Human Development (CIDESD), 5000-801 Vila Real, Portugal; anaconceicao@esdrm.ipsantarem.pt (A.C.); hlouro@esdrm.ipsantarem.pt (H.L.); ajsilva@utad.pt (A.J.S.); mario.costa@ipg.pt (M.J.C.); 5Department of Sport Sciences, Sport Sciences School of Rio Maior, 2040-413 Rio Maior, Portugal; 6Department of Sports Sciences, University of Trás-os-Montes and Alto Douro, 5000-801 Vila Real, Portugal; 7Department of Sports Sciences, Polytechnic Institute of Guarda, 6300-559 Guarda, Portugal

**Keywords:** swimming, isometric strength, muscle balance, shoulder rotators

## Abstract

The purpose of this study was to analyze the acute effects of a standardized water training session on the shoulder rotators strength and balance in age group swimmers, in order to understand whether a muscle-strengthening workout immediately after the water training is appropriate. A repeated measures design was implemented with two measurements performed before and after a standardized swim session. 127 participants were assembled in male (*n* = 72; age: 16.28 ± 1.55 years, height: 174.15 ± 7.89 cm, weight: 63.97 ± 6.51 kg) and female (*n* = 55; age: 15.29 ± 1.28 years, height: 163.03 ± 7.19 cm, weight: 52.72 ± 5.48 kg) cohorts. The isometric torque of the shoulder internal (IR) and external (ER) rotators, as well as the ER/IR ratios, were assessed using a hand-held dynamometer. Paired sample t-tests and effect sizes (Cohen’s d) were used (*p* ≤ 0.05). No significant differences were found on the shoulder rotators strength or balance in males after training. Females exhibited unchanged strength values after practice, but there was a considerable decrease in the shoulder rotators balance of the non-dominant limb (*p* < 0.01 *d* = 0.366). This indicates that a single practice seems not to affect the shoulders strength and balance of adolescent swimmers, but this can be a gender specific phenomenon. While muscle-strengthening workout after the water session may be appropriate for males, it can be questionable regarding females. Swimming coaches should regularly assess shoulder strength levels in order to individually identify swimmers who may or may not be able to practice muscle strengthening after the water training.

## 1. Introduction

The shoulder rotator muscles play a critical role in providing stability and mobility to the glenohumeral joint and shoulder joint complex [1]. In competitive swimming, the propulsive forces responsible for the total body displacement are produced mainly by the upper limbs through the arm adduction and shoulder internal rotation [2]. As such, the swimmers are classified as overhead athletes [3] because high levels of stress are installed in the upper body sections.

There is evidence that water training induces shoulder muscle imbalances [4]. The shoulder adductor and the internal rotator (IR) show a tendency to become proportionally stronger when compared to their antagonists. In addition, higher volumes of swimming training are associated with shoulder pain and injury [5]. Without the addition of preventive measures, this can lead to an acute or chronic injury process [6].

Shoulder strengthening programs are part of the training season and require detailed planning [3,7]. Evidence suggests that dryland workouts must be maintained throughout the entire sports season, otherwise the effects of detraining are felt [4]. Few studies have focused on shoulder rotator strengthening in competitive swimmers. Batalha et al. [8] found that a 10-week dryland training program reduced muscle imbalance and fatigue. Kluemper et al. [2] also reported increases in the shoulder rotators strength after a 6-week training program with consequent postural improvements. So, regular shoulder strength workouts are essential to maintain the integrity and longevity of the swimmers’ glenohumeral joint [8].

Most times dryland strength training and in-water training are applied within the same training session. Coaches regularly assign dryland strength training before the swimming practice due to time-consuming issues [3,9]. However, it remains unclear when these workouts should be implemented and the impact an in-water session has on shoulder fatigue. It is not consensual if a single water session will compromise the shoulder joint and muscles, and what would be the degree of fatigue installed. The increasing fatigue in shoulder muscles, especially in the rotator cuff group, was identified as a possible cause of shoulder dysfunction [10] and was associated with performance decrements and a higher risk of injury [11]. Regarding the performance of muscle strengthening programs before the water training, Batalha et al. [9] evaluated the acute effects on the shoulder rotator strength. They concluded that shoulder rotator endurance and balance do not seem to be impaired after undergoing a shoulder rotator injury-prevention training program. However, to the best of our knowledge, there are no studies that assess the effects water training may have on the strength and balance of the shoulder rotators. Hence, it is crucial to understand whether it would be possible to implement a shoulder rotator strengthening program after swimming practice. Thus, the aim of this study was to analyze the acute effects of a standardized water training session on the strength and balance of the shoulder rotators in adolescent swimmers. It was hypothesized that a swimming training session would significantly reduce the shoulder rotator strength levels and muscle balance, limiting the performance of the dryland training afterward.

## 2. Materials and Methods

### 2.1. Participants

One hundred and twenty-seven national-level adolescent swimmers participated in this study. All participants were recruited from the clubs that agreed to participate in the study. Seventy two participants were males (age: 16.28 ± 1.55 years, height: 174.15 ± 7.89 cm, weight: 63.97 ± 6.51 kg, training/week: 6.75 ± 0.86 sessions, training time/day: 126 ± 26.39 min), and 55 were females (age: 15.29 ± 1.28 years, height: 163.03 ± 7.19 cm, weight: 52.72 ± 5.48 kg, training/week: 6.52 ± 0.57 sessions, training time/day: 115 ± 16.23 min) who met the following inclusion criteria: (i) do not have any clinical history of upper limb disorders; (ii) compete at the national level; and (iii) have a minimum of 10 h of training per week. The main goals of the study were explained to all participants and their legal guardians, who signed an informed consent allowing their participation. This research was approved by the ethics committee of the seeding institution. The research was undertaken in compliance with the Declaration of Helsinki and the international principles governing research on humans and animals.

### 2.2. Procedures

Before the data collection, all participants completed a questionnaire that included questions on hand dominance, shoulder injury, pain, and swimming training frequency. All subjects performed a 5-min shoulder warm-up with articular mobility and resistance tubing with the same directions used for testing. The IR and ER cuff strength data were collected during isometric actions performed with the microfet 2™ Digital Handheld Muscle Tester (Hoggan health, Draper, UT, USA), which is an accurate, portable Force Evaluation and Testing device, with a sample frequency of 10 sample/second. It is a modern adaptation of the time-tested art of hands-on manual muscle testing. The hand-held dynamometer (HHD) has wireless capability, is battery operated, and is ergonomically designed to fit comfortably in the palm of the hand. The reliability of this device to measure ER/IR ratios has already been documented [12]. An experienced tester with more than 10 years of experience performing muscle strength measurement with an HHD conducted all the tests.

Shoulder ER and IR strength were evaluated with the exact same methodology before and after a standardized swimming training. Tests were performed bilaterally in prone position with 90° of shoulder abduction and 0° of rotation with the elbow flexed to 90° (Figure 1). This position was considered the most suitable to replicate the arm position during the stroke, while also being repeatable [13,14]. Additionally, it has been used in previous studies within the same topic [3,15]. The order of testing position, sides (dominant, non-dominant), and motions (internal rotation, external rotation) was randomized. The tester stabilized the humerus distally against the stretcher, and the participants used the opposite arm to grasp it next to the test table for support. The HHD was placed just proximal to the ulnar styloid process on the anterior surface of the forearm to assess the IR strength. To assess the ER strength, the HHD was positioned using the same anatomical landmarks but on the posterior surface of the forearm. After positioning, the participants performed a warm-up protocol that consisted of 2 submaximal and 1 maximal isometric contraction in each direction, separated by 30 s of rest [14]. The maximal IR and ER isometric strength was evaluated with two repetitions for each shoulder and each rotation (2 × IR and 2 × ER), using a make-type test in which the participants were instructed to slowly produce and sustain a full isometric contraction (five seconds) of the involved muscle group until they were told to relax. The maximum value recorded from the two repetitions of each test session was used for analysis. In order to analyze muscle balance, the ER/IR ratio [(ER/IR) ×100] was calculated [16]. All tests had a resting period of 10 s between each repetition and 60 s between each strength test. During the entire period of force production, the tester verbally motivated the participants.

A “standard” swimming training session was performed between measurements. The training session was carried out according to the recommendations of the Portuguese Swimming Federation regarding the activities of the youth national squads participating in international competitions [17]. The total volume of the training session was 4600 m, including: (i) 900 m of warming-up tasks with low-intensity bouts; (ii) 800 m of technical training with technical drills; (iii) 400 m of velocity tasks with sprint bouts; (iv) 1000 m of aerobic capacity bouts; (v) 600 m of aerobic power bouts; and (vi) 900 m of recovery sets with low-intensity tasks. The session was previously sent to all coaches for approval and to be part of the training unit within each testing day.

### 2.3. Statistical Analysis

Assumptions of statistical tests such as normal distribution (Shapiro-Wilk test, *p* > 0.05) and sphericity (Mauchly test, *p* > 0.05) of data were checked as appropriate. All the parameters were normally distributed. Paired sample t-tests were conducted to compare the means of the shoulder strength, before and after the water training. In addition to the *p* values, the researchers provided detailed statistics, including the mean, the mean differences, and 95% confidence interval, to better depict the changes between the two testing points. Effect sizes were calculated using Cohen’s d and interpreted as: null if <0.2, small if 0.2 to <0.6, medium if 0.6 to <1.2, and large if ≥1.2 [18]. All analyses were performed with SPSS (version 23.0; SPSS Inc, Chicago, IL, USA), and the significance level was set at *p* ≤ 0.05 for all tests.

## 3. Results

Table 1 shows the strength and balance measures of the overall sample in pre and post-test. Although slight decreases in the strength and muscle balance are seen, those are not significant between both time points. Even though the result of the ER/RI ratio of the non-dominant shoulder was near the cut-off value for significance (*p* = 0.050; *d* = −0.175) this should be highlighted as a considerable reduction.

Table 2 presents the shoulder rotators strength of male swimmers. Since there were no significant differences between testing points, there were no acute effects of aquatic training in this group. All the effect size values were under 0.2, indicating a null effect. Thus, there seems to be no issue carrying out shoulder-strengthening training programs following the water training with male swimmers.

Table 3 shows the results for female swimmers, revealing some differences compared to their male counterparts. Although there were unchanged strength values between testing points, there was a considerable decrease in the balance of the shoulder rotators of the non-dominant limb after practice. These results suggest that there should be some caution when conducting strengthening programs after aquatic training with female swimmers.

The analysis of the different effect sizes in the distinct measurements is presented in Figure 2. The acute effect of the water training is related to a decrease in the ER/IR ratio mostly in the non-dominant shoulder of female swimmers (*p* < 0.01, *d* = 0.366).

## 4. Discussion

The aim of this study was to analyze the acute effects of a standardized water training session on the shoulder rotator strength and balance in adolescent competitive swimmers. The results do not fully confirm the initial hypothesis that an in-water training session is enough to reduce the shoulder rotator strength levels and muscle balance. In fact, this seems to be a gender specific phenomenon. While the male swimmers revealed no acute effect after the water session, the female swimmers showed a significant increase in muscle imbalance in the non-dominant shoulder.

The comparative analysis of the strength levels between ER and IR is consistent in the total sample and between genders. This allows us to confirm that the muscle groups responsible for the internal rotation are in fact the strongest, and this is in line with previous studies that included swimmers [3,9], athletes from other sports [6,19], and sedentary people [20]. This can be explained by the size and number of the muscles around the glenohumeral joint. The muscular groups which produce the IR are not only greater in number, but are also anatomically larger and naturally stronger than their antagonists [21].

Additionally, in relation to IR and ER strength, it is a fact that shoulder muscles fatigue induces sensory motor perturbations that consequently alter the kinematics of the shoulder joint, overloading different structures in order to maintain the level of performance [22]. It is still unclear whether rotator cuff-specific muscle fatigue or general scapular muscle fatigue has the greatest influence on upper limb malfunction; however, it is a fact that there is a difference in the behavior and reaction regarding muscle fatigue around the shoulder joint, IRs appearing to be more fatigue-resistant than ERs in both physically-active people [23] and sedentary ones [20]. Our results do not fully support this statement since, in general, the IR strength values showed greater reductions compared to the ER, although with no significant values. In the analysis of the total sample results (Table 1), we can see that, despite there being no significant differences between the pre and post aquatic training, the reductions in the IRs strength levels are higher than those of the ERs (in the non-dominant shoulder there was no reduction). These results may point out that, contrary to what happens in other sports, swimming induces a greater degree of fatigue in the IR, which can be explained by the predominant internal-rotator forces that occur during pull-out and recovery [24]. During all stroke phases, the subscapularis muscle is always active, stabilizing the glenohumeral joint, because of the repetitive internal rotator forces [25]. It should be emphasized that IRs strength deficits may affect stroke dynamics and should be considered an important injury risk factor for swimmers [26].

When analyzing the results of the ER and IR strength values by gender, there was no acute effect in any of the variables after performing the aquatic training, but it showed some specific trends. In male swimmers, the strength values slightly decreased in the post-test, mostly in the IRs. Contrarily, in females, there was only a reduction in the IR values for the non-dominant shoulder, which can be seen as an alarming cue. Overall, the results for both genders are in agreement with the study by Matthews et al. [27]. The authors reported no significant strength differences between pre- and post-fatigue in the internal and external rotators of both shoulders. The changes were exclusively reductions in the stroke length, and in the IR and ER range of motion on both shoulders. More recently, Yoma et al. [24] observed that shoulder rotation isometric peak torque decreased immediately after a high-intensity training session but remained unchanged after a low-intensity session. These results highlight the importance of evaluating the strength values of the shoulder rotators, while controlling the training intensity. In this study, we did not perform an effective control of intensity, so no conclusions can be drawn. However, considering the characteristics of the swimming tasks (pointing to a medium/low intensity effort), we may argue that the intensity was not enough to induce fatigue as it happened in previous studies.

Regarding the shoulder rotators balance (ER/IR ratios), numerous studies reported that swimmers had greater IRs strength because of the repetitive concentric actions required during the propulsive phases of the swimming strokes [3,7]. In contrast, ER strength is weaker, which often leads to shoulder muscle imbalances. The ER/IR ratio assessment can be a useful measure to identify muscle imbalances in the swimmers’ shoulders, and it is also associated with shoulder injuries [28], and scapular dyskinesis [29]. Previous normative data consider ER/IR ratios between 66 and 75% to be adequate [28,30]. Values below this threshold are commonly associated with muscle imbalance and instability in the glenohumeral joint [23]. In this study, the ER/IR ratios are within the considered normal standards for a healthy joint [28,30], varying between 70.22% ± 12.80% and 79.50% ± 14.58% in male swimmers, and between 77.07% ± 9.64% and 89.69 ± 26.78% in female swimmers in pre- and post-tests. In addition, these results are in line with previous studies that included swimmers [12], in which the authors carried out a reliability study, with the same evaluation technique, with ratios between 78.71% ± 9.36 and 81.81% ± 10.24%. The similarity of the results remained when compared to the study by Batalha et al. [9], but in this case the authors evaluated swimmers with an isokinetic dynamometer. The ratios were between 72.31% ± 15.66% and 77.37% ± 16.40%. On the other hand, Riemann et al. [14], in order to compare different evaluation positions with a handheld dynamometer, used a position and methodology similar to the one used in the current study. The ER/IR ratios presented were considerably higher (between 90% and 92% in boys, and 98 and 99% in girls). However, the sample was composed of healthy non-sports individuals, which may prove that swimmers have greater shoulder muscle imbalances when compared to non-sports individuals [20].

When we analyze the results of the ER/IR ratios, we can see that in male swimmers there was no acute effect of aquatic training on the shoulder rotators balance, since the ratio values are practically identical in pre- and post-training. In female swimmers, there was a slight reduction in the ratio in the dominant limb, although without a significant result. However, in the non-dominant shoulder there was a considerable acute effect of the water training (*p* = 0.009; *d* = 0.366). While this significant reduction is within the normal thresholds [30], coaches must be aware of it. Since the muscular strength balance between agonist and antagonist muscles is crucial for joint stability and ensures a dynamic centralization of the humeral head [29], this can be even more compromised if a strengthening training is to be implemented after a swim session.

Some limitations should be considered for further discussion: (i) no strict control of the training intensity may have compromised the strength response from swimmers; although the session was carefully planned, the effort during each training task was dependent on the athlete (ii) the assessment of strength deficits after the swimming practice was limited to the shoulder group, not allowing the study of those effects in other muscles in action.

## 5. Conclusions

The results of this study indicate that a single practice seems not to affect shoulders strength and balance of adolescent swimmers, but this can be a gender specific phenomenon. While a muscle-strengthening workout after the water session may be appropriate for males, this can be questionable for females, since the shoulder rotators’ muscular balance, mostly of the non-dominant limbs, was compromised after training.

We believe these results have important practical implications for swimmers and coaches. In fact, swimming coaches should be careful if they intend to carry out shoulder-strengthening programs immediately after swim practice. It should be important to identify deficits in post swim rotation strength, serving as a practical way to reduce the athlete’s susceptibility to shoulder injury. In addition, an individualized regular exercise program to improve shoulder rotation strength should be performed to minimize the post swimming adaptations. Finally, we believe that if swimming coaches regularly assess strength levels, they can easily find a method to check individual trends and see who will be able or not to maintain the strengthening workouts after the water training sessions.

## Figures and Tables

**Figure 1 ijerph-18-08109-f001:**
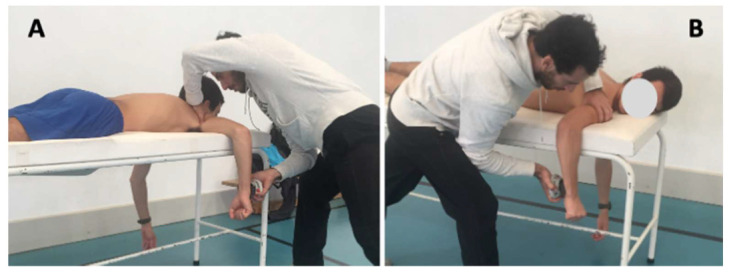
Shoulder external and internal rotation measurements. (**A**)—Shoulder external rotation measurement; (**B**)—Shoulder internal rotation measurement.

**Figure 2 ijerph-18-08109-f002:**
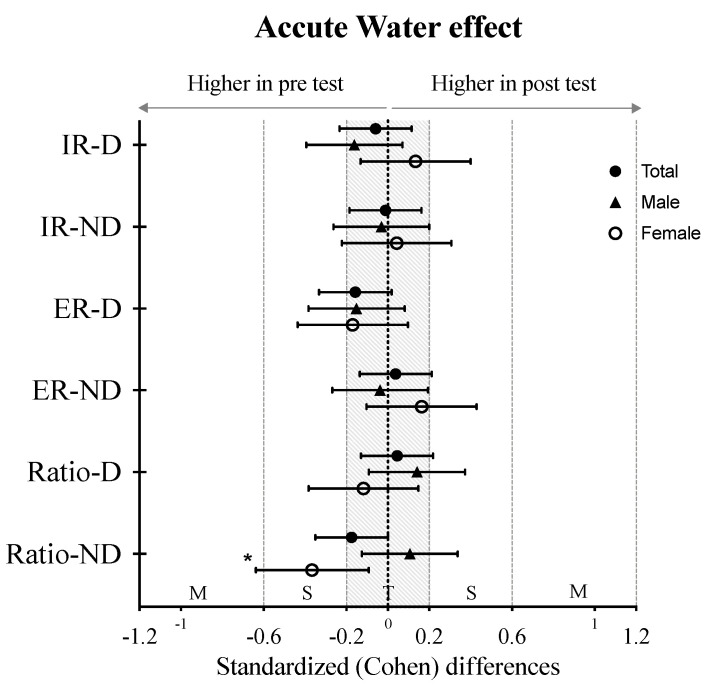
Cohen’s d values (Mean ± SD) of pre and post water training. Paired-sample *t*-test, * *p* ≤ 0.05.

**Table 1 ijerph-18-08109-t001:** Maximal isometric strength of internal and external, dominant and non-dominant shoulders, before and after training.

	Before Training *N* = 127	After Training *N* = 127	Mean Difference (95% CI)	*p*
IR_D (N)	169.40 ± 68.6	167.84 ± 45.3	−1.49 (−5.89, 2.92)	0.505
IR_ND (N)	172.60 ± 50.1	168.40 ± 50.9	−4.20 (−8.62, 0.48)	0.079
ER_D (N)	131.50 ± 35.3	130.63 ± 33.6	−0.87 (−4.08, 3.59)	0.901
ER_ND (N)	120.46 ± 28.88	121.10 ± 29.21	0.64 (−2.33, 3.61)	0.672
Ratio_D (%)	77.90 ± 11.71	78.50 ± 12.75	0.55 (−1.6, 2.71)	0.612
Ratio_ND (%)	78.65 ± 22.22	74.57 ± 13.84	−4.08 (−8.15, 0)	0.050

*p*-values for Paired sample *t*-tests; IR—Internal Rotation; ER—External Rotation; D—Dominant; ND—Non-Dominant.

**Table 2 ijerph-18-08109-t002:** Maximal isometric strength of internal and external, dominant and non-dominant shoulders, before and after training.

	Before Training *N* = 72	After Training *N* = 72	Mean Difference (95% CI)	*p*
IR_D (N)	196.30 ± 37.5	191.69 ± 40.60	−4.61 (−11.27, 2.06)	0.173
IR_ND (N)	200.83 ± 42.88	196.43 ± 45.18	−4.4 (−11.22, 2.42)	0.202
ER_D (N)	151.15 ± 30.54	150.31 ± 32.98	−0.84 (−7.16, 5.48)	0.791
ER_ND (N)	137.73 ± 23.9	137.04 ± 24.63	−0.69 (−5.04, 3.67)	0.755
Ratio_D (%)	77.60 ± 11.72	79.50 ± 14.58	1.9 (−1.27, 5.07)	0.236
Ratio_ND (%)	70.22 ± 12.80	71.62 ± 13.83	1.4 (−1.69, 4.49)	0.369

*p*-values for Paired sample *t*-tests; IR—Internal Rotation; ER—External Rotation; D—Dominant; ND—Non-Dominant.

**Table 3 ijerph-18-08109-t003:** Maximal isometric strength of internal and external, dominant and non-dominant shoulders, before and after training.

	Before Training *N* = 55	After Training *N* = 55	Mean Difference (95% CI)	*p*
IR_D (N)	134.03 ± 25.39	136.62 ± 29.33	2.6 (−2.65, 7.85)	0.326
IR_ND (N)	134.02 ± 29.43	130.39 ± 29.01	−3.64 (−9.42, 2.14)	0.212
ER_D (N)	103.61 ± 19.24	104.16 ± 21.65	0.55 (−2.88, 3.97)	0.750
ER_ND (N)	97.86 ± 16.59	100.23 ± 20.08	2.37 (−1.55, 6.29)	0.231
Ratio_D (%)	78.29 ± 11.66	77.07 ± 9.64	−1.21 (−3.99, 1.57)	0.386
Ratio_ND (%)	89.69 ± 26.78	78.45 ± 12.98	−11.24 (−19.53, −2.96)	0.009

*p*-values for Paired sample *t*-tests; IR—Internal Rotation; ER—External Rotation; D—Dominant; ND—Non-Dominant.

## Data Availability

The data presented in this study are available on request from the corresponding author. The data are not publicly available due to privacy restrictions.

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
