# Peer review of "The Acute Effects of a Swimming Session on the Shoulder Rotators Strength and Balance of Age Group Swimmers"

_ijerph, 2021, doi:10.3390/ijerph18158109_

Round 1

Reviewer 1 Report

Dear author, I like your paper because it puts a focus on a bis issue in swimming. I would like you to extend the results session. That makes the discussion clearer. 

Author Response

Thank you for your valuable comment. Considerable changes have been made to make it clear.

Reviewer 2 Report

The authors of the present paper evaluated the effects of a swim training on shoulder-muscle forces. The found no relevant reduction in males, whereas this was not clear in females.

The paper is generally well written. Although the topic is very specific, the results may be of relevance for readers in this specific field. I have three comments to address:

  1. Please add a short information on how participants were recruited.
  2. I miss the statistic section. I think it should be easy to add this.
  3. Please provide details of the so called "standardized training session" (e.g., intensity, duration, TRIMP or energy expenditure).

Author Response

We would like to thank to all the reviewers for your comments. They actually contribute to improve the quality of the document. We have made considerable changes in the document. However, we would like to clarify some points that could have been misinterpreted.

1. Please add a short information on how participants were recruited.

New information was added based on your comment.

2. I miss the statistic section. I think it should be easy to add this.

We agree with your comment. Somehow when uploading the final version of the manuscript we didn’t checked that statistics were missing. So, the statistical analysis section has now been added for clarity.

3. Please provide details of the so called "standardized training session" (e.g., intensity, duration, TRIMP or energy expenditure).

The standardized training session was described deeper on page 3and 4. We understand your point and try to dissect how training sets were applied in terms of intensity and recovery between bouts. Please keep in mind that we tried to design a training session that would allow all swimmers to complete and easy to apply for coaches. For that, we followed the guidelines proposed by the Portuguese Swimming Federation for training implementation (Marinho et al. 2020). Although energy expenditure is important, we not used any resource to get oxygen uptake since that was not the aim of this study. At the end we hope that description would meet your requirements now.

Reviewer 3 Report

I would like to congratulate the authors for the study carried out.

My comments to the authors are:

Line 77: A sentence should not begin with numbers or acronyms. Authors must write the whole word

The authors took into account in the selection criteria of the sample if the swimmers had any limitation to movement in the shoulder?

The authors should explain in more detail the recruitment process of the sample.

Photos should be in color

The authors should add a section explaining the statistical analysis that they have carried out in the study.

Lime 128: Authors must delete the sentence

Line 159: there is a misspelled word “analyse”

Researchers must conduct a more detailed analysis of their study. A comparison between arms, between dominant and non-dominant arm, ... It is an interesting topic but the authors do not work it well.

Authors should improve the statistical analysis and results section

The conclusion is too long. Authors should synthesize the clinical finding of their study in 2-3 sentences

References: The style of the references is correct.

Author Response

We would like to thank to all the reviewers for your comments. They actually contribute to improve the quality of the document. We have made considerable changes in the document. However, we would like to clarify some points that could have been misinterpreted.

Line 77: A sentence should not begin with numbers or acronyms. Authors must write the whole word

We totally agree with your comment. The change was made throughout the manuscript.

The authors took into account in the selection criteria of the sample if the swimmers had any limitation to movement in the shoulder?

We appreciate your comment. However, in the "Materials and methods" section, we think that the inclusion criteria is clear: "i) not have any clinical history of upper limb disorders". So, here we assumed that all the included swimmers did not show any shoulder movement limitation.

The authors should explain in more detail the recruitment process of the sample.

Smooth changes were made in the “Participants” section.

Photos should be in color

The figure has been changed accordingly

The authors should add a section explaining the statistical analysis that they have carried out in the study.

We agree with your comment. Somehow when uploading the final version of the manuscript we didn’t checked that statistics were missing. So, the statistical analysis section has now been added for clarity.

Lime 128: Authors must delete the sentence

The sentence has been deleted. Thank you for your contribution.

Line 159: there is a misspelled word “analyse”

We appreciate it. The word has been corrected throughout the document.

Researchers must conduct a more detailed analysis of their study. A comparison between arms, between dominant and non-dominant arm, ... It is an interesting topic but the authors do not work it well.

We understand the comment, which we appreciate in advance. However, for the purpose of this study (to analyze the acute effects of a standardized water training session on the shoulder rotators strength and balance in adolescent swimmers), it does not seem relevant to make a comparison between dominant and non-dominant shoulder.

Previous investigations (Batalha et al. 2012; Batalha et al. 2013) showed no differences in swimmers indicating the bilateral movements as the main cause for that. The problem exists in the unilateral ratios, which assess external and internal rotators of the same shoulder. Previous data (Batalha et al. 2015) reported that shoulder rotators imbalances may lead to future injuries. So, in this study we tried to avoid presenting redundant data and explore new findings. Your comment seems valid, but it can be considered in a near future, probably adding other measures in parallel for a better understanding.

Batalha, N. Raimundo, A. Carus, P. Fernandes, O. Marinho, D. Silva, A. (2012). Shoulder rotator isokinetic strength profile in young swimmers. Brazilian Journal of Kinanthropometry and Human Performance, 14(5), 545-553.

Batalha, N. Raimundo, A. Carus, P. Barbosa, T. Silva, A. (2013). Shoulder rotator cuff balance, strength and endurance in young swimmers during a competitive season. Journal of strength and conditioning research. 27(9), 2562-2568.

Batalha, N., Marmeleira, J., Garrido, N., & Silva, A. J. (2015). Does a water-training macrocycle really create imbalances in swimmers’ shoulder rotator muscles? European Journal of Sport Science, 15(2), 167–172. doi:10.1080/17461391.2014.908957

Authors should improve the statistical analysis and results section

Changes have been made in both sections.

The conclusion is too long. Authors should synthesize the clinical finding of their study in 2-3 sentences

We appreciate your comment. Few changes have been made in the conclusions section. However, please keep in mind that there was also a need to respond to the comments of the other reviewers, who requested a deeper description on the practical implications for swimmers and coaches. So, we appeal to your understanding.

Reviewer 4 Report

The paper is interesting and within the scope of the journal. However, there are some changes that should be made to improve the quality of the paper. 1. Please revise the English spelling. 2. Please rewrite the introduction to highlight the rational of the study. 3. Please address more clearly the aims of the paper and the hypothesis. 4. Please add more data on the sample. 5. Please define the intensity levels of the training session. Is it a typical high intensity training session? 6. Please address the discussion according to the literature review and the introduction. 7. Please add some more practical implications to the swimmers and coaches.

Author Response

We would like to thank to all the reviewers for your comments. They actually contribute to improve the quality of the document. We have made considerable changes in the document. However, we would like to clarify some points that could have been misinterpreted.

The paper is interesting and within the scope of the journal. However, there are some changes that should be made to improve the quality of the paper.

  1. Please revise the English spelling.

The entire document was reviewed by a company specialized in English proofreading (editorial certificate attached).

  1. Please rewrite the introduction to highlight the rational of the study.

We understand your point and try as much as we can add new information for clarity. We hope that both rational and research gap are clear for the reader.

  1. Please address more clearly the aims of the paper and the hypothesis.

We appreciate your comment. We believe that the sentences presented on the page 2 clearly indicate the aim of the study and the formulated hypothesis:

“… the aim of this study was to analyze the acute effects of a standardized water training session on the strength and balance of the shoulder rotators in adolescent swimmers. It was hypothesized that a swimming training session would significantly reduce the shoulder rotator strength levels and muscle balance, limiting the performance of the dryland training afterward.”

  1. Please add more data on the sample.

Smooth changes were made and new info was added in the participants’ section. But, we consider that the sample characteristics described (age, height, weight, number of training sessions per week and the average training time per session) are enough for that purpose. Plus, this description is in agreement with the standards proposed for studies quality in terms of sample description (item evaluated in most of systematic reviews) and has the same information than most of the studies published in IJERPH and MDPI journals. Here we hope your understanding.

  1. Please define the intensity levels of the training session. Is it a typical high intensity training session?

The standardized training session was described deeper on pages 3-4. We understand your point and tried to dissect how training sets were applied in terms of intensity and recovery between bouts. Please keep in mind that we tried to design a training session that would allow all swimmers to complete and easy to apply for coaches. For that, we followed the guidelines proposed by the Portuguese Swimming Federation for training implementation (Marinho et al. 2020). At the end we hope that description in this new version would meet your requirements now.

  1. Please address the discussion according to the literature review and the introduction.

We understand your point and tried as much as we can add new information for clarity, especially in the conclusion section to try to connect more with the discussion and the main findings of the paper. Thank you very much for your interest.

  1. Please add some more practical implications to the swimmers and coaches.

Few changes have been made in the conclusions section. However, please keep in mind that there was also a need to respond to the comments of the other reviewers. Some asked for a reduction in this section and others to explore deeply. We tried to consolidate all the reviewers comments and give a short and fairly and practical conclusion.

Reviewer 5 Report

The presented work has interesting assumptions. Matches the journal's profile. I recommend the following comments to the authors.
1. The results are interesting. The data was described correctly, but there is no more detailed description of the statistical tools used and the reasons for using these particular statistics.
2. The conclusions are interesting, but they can be reformulated in an even more appropriate direction for swimming coaches.
3. No need to present in the text Figure 1 - Measurements of external and internal arm rotation as the whole procedure is sufficiently described in the text. 

Author Response

We would like to thank to all the reviewers for your comments. They actually contribute to improve the quality of the document. We have made considerable changes in the document. However, we would like to clarify some points that could have been misinterpreted. The presented work has interesting assumptions. Matches the journal's profile. I recommend the following comments to the authors.

1. The results are interesting. The data was described correctly, but there is no more detailed description of the statistical tools used and the reasons for using these particular statistics.

Somehow when uploading the final version of the manuscript we didn’t checked that statistics were missing. So, the statistical analysis section has now been added for clarity.
2. The conclusions are interesting, but they can be reformulated in an even more appropriate direction for swimming coaches.

We appreciate your comment. The conclusions section has been completely restructured, highlighting the practical implications for coaches and swimmers.

  1. No need to present in the text Figure 1 - Measurements of external and internal arm rotation as the whole procedure is sufficiently described in the text. 

We understand your comment and appreciate it. However, in line with another reviewer comments, we decided to keep figure 1 and change it to a colored picture.

Round 2

Reviewer 3 Report

The authors have gone to great lengths to improve the paper. They have answered all my comments